# Feline Demodicosis Case Report—First Molecular Characterization of *Demodex* Mites in Romania

**DOI:** 10.3390/pathogens10111474

**Published:** 2021-11-12

**Authors:** Marius Stelian Ilie, Mirela Imre, Simona Giubega, Iasmina Luca, Tiana Florea, Sorin Morariu

**Affiliations:** Department of Parasitology and Dermatology, Banat‘s University of Agricultural Sciences and Veterinary Medicine “King Michael I of Romania” from Timisoara, no. 119, Calea Aradului, 300645 Timisoara, Romania; mirela.imre@usab-tm.ro (M.I.); simonagiubega@gmail.com (S.G.); iasmina.luca@usab-tm.ro (I.L.); tijana.florea@usab-tm.ro (T.F.); sorin.morariu@fmvt.ro (S.M.)

**Keywords:** cat, PCR, *Demodex cati*, Romania, fluralaner, moxidectin

## Abstract

Cat demodicosis is uncommon to rare, and is caused by *Demodex cati*, *Demodex gatoi* and another unnamed species. The investigated patient was a mix-breed, 10-year-old feline with no dermatological history. Alopecia, erythema, minor erosions and ulcerations and crusts, associated with pruritus and self-trauma, were observed on the head. Dark, agglutinated cerumen was also present in the external ear canal. The agent causing the skin condition in the feline patient was identified as being a *Demodex* genus mite, based on the specific, morphological characteristics noticed upon the microscopic examination of deep skin scrapes. Biological samples were collected from the patient with to perform a PCR assay for clear species-determination and morphological assessment. PCR amplification of DNA extracted from the *Demodex* mites produced a single band of ~330 bp, indicating the presence of the *D. cati* species. The acaricidal treatment consisted of topical treatment using a fluralaner and moxidectin-based spot-on. Upon follow-up appointments, scheduled three times at a monthly interval, the patient failed to provide a positive result upon deep skin scrapes. The negative scrapes were also accompanied by the complete resolution of the existing lesions. In conclusion, this is the first molecular study to highlight the presence of *Demodex cati* within the feline population of Romania, and the fluralaner-moxidectin spot-on therapy has led to a complete recovery of the feline patient affected by feline demodicosis.

## 1. Introduction

Demodicosis is a skin condition caused by *Demodex* genus mites (Acariformes: Demodecidae). Feline demodicosis is an uncommon to rare condition, and it is caused by three species of *Demodex* mites (*D. cati*, *D. gatoi* and one species that, despite being mentioned in several research papers, has yet to receive a name) [1].

These mites show high host specificity [2] and their location can vary according to species. Thus, *D. gatoi* is identified in the epidermal layer, as opposed to *D. cati*, which is found within the hair follicle [3].

This condition has been mainly diagnosed in immune-suppressed cats, such as cats suffering from chronic diseases like FIV, diabetes mellitus, squamous cells carcinoma or chronic breathing disorders [4,5].

The most frequent locations identified in feline patients include the head (ears and periocular region), and the dorsal area. Feline demodicosis is generally manifested through well-defined alopecia, erythema and crusts, all of which are lesions that could evolve into pruritic, bacterial complications [6,7].

## 2. Materials and Methods

### 2.1. Case Report

#### 2.1.1. Clinical Exam

The investigated patient was a mixed-breed, 10-year-old cat with no dermatological history. The cat was adopted by the owner at the age of 3 months, and it was spayed after having 2 consecutive births (2 years of age). Thirty days before the consulting, in November 2019, the cat had disappeared from home. Upon her return, the owner noticed symptoms such as excessive scratching and hair loss, associated with weight loss as well as increased sensitivity to external stress factors.

Alopecia, erythema, minor erosions and ulcerations and crusts, associated with pruritus and self-trauma, were observed on the head (Figure 1a–c). Dark, agglutinated cerumen was present in the external ear canal.

#### 2.1.2. Diagnostic

The main differential diagnoses included a ringworm infection, *Malassezia* dermatitis, bacterial infections and mite infestations. Following clinical examination, we have proceeded to perform a Wood’s lamp test, a trichogram, ear-cytology, otoscopy and a skin scrape.

Several deep skin scrapes were performed from the lesion margins, the obtained material was mixed in mineral oil, a coverslip was applied, and the samples were examined directly, using the 10x magnification objective of a microscope.

Measurements were also taken, in order to establish the specific sizes of the mites’ body segments (gnathosoma, podosoma and opisthosoma).

Otoscopy and cytological examinations of the ear secretions were performed. Subsequent bacterial culture and antibiogram were also performed, Due to the excessive weight loss of the animal, we also performed a quick test (FIV Ab/FeLV Ag—Vet Expert) in order to identify the possible presence of these debilitating viruses, based on the identification of serum, blood or plasma antibodies. The first follow-up was scheduled within 3 weeks after the initial consult and the next follow-ups were scheduled on a monthly basis, for four consecutive months. During every follow-up visit, we performed skin scrapes, and we established the clinical evolution, following treatment.

### 2.2. PCR Protocol

In order to achieve clear species-determination, we collected biological material for PCR.

DNA was extracted from the skin scrapings, using a commercial kit, according to the manufacturer’s protocol for tissue samples (ISOLATE II Genomic DNA Kit, Bioline^®^, Cincinnati, OH, USA). It was then amplified by PCR, as previously described by Frank et al. [8], using forward primer ACTGTGCTAAGGTAGCGAAGTCA and reverse primer TCAAAAGCCAACATCGAG to amplify 16S rRNA DNA, 2 µL Extracted DNA and MyTaq Red Mix (Bioline^®^). PCR amplification parameters were 95°C for 90 s, followed by 35 cycles of 55°C for 30 s, 68°C for 120 s and 94°C for 30 s, with a final cycle consisting of 55°C for 30 s and 68°C for 5 min. The PCR product was visualized on 1.5 % agarose gel, stained with Midori Green™ (Nippon Genetics^®^; Europe Gmbh, Düren, Germany), cleaned with Isolate II PCR and Gel Kit (Bioline^®^) and sequenced using the forward and reverse PCR primers at Macrogen Europe® Company (Amsterdam, the Netherlands) (Appendix A).

BLAST was used to compare the sequence with all entries available in GenBank.

### 2.3. Therapeutic Protocol

The acaricidal treatment consisted of topical treatment using a fluralaner and moxidectin based pipette (Bravecto plus^®^, MSD, Kenilworth, NJ.USA). The dose was 250 mg/animal weighing between 2.8–6.25 kg, administered monthly (4 administrations overall). In order to improve the immune function, we recommended the use of RX immuno support^®^ (Rx vitamins, Elmsford, NY, USA) capsules, given *per os* (1x1/day, for 10 days, repeated after 10 days).

The recommended treatment of the bacterial otitis consisted in daily cleansing of the external ear canal using Epiotic^®^ (Virbac, Carros, France), a salicylic acid based solution, followed by instillation of 3-5 drops/ear of Aurizon® (Virbac)—marbofloxacin, clotrimazole, dexamethasone acetate, twice/day for 15 days.

## 3. Results and Discussion

Following otoscopy and cytological examination of the ear secretions, we ruled out *Otodectes cynotis* infestation and established the diagnostic of bacterial otitis. The following bacterial strains were identified through the subsequent bacterial culture: *Staphylococcus intermedius*, *Pseudomonas aeruginosa*, *beta-hemolytic Streptococci*. An antibiogram was also performed, highlighting sensitivity of the identified pathogens to chloramphenicol (++++), ciprofloxacin, tobramycin, amikacin (+++) and resistance to trimethoprim, lincomycin, spectinomycin (-).

All the scrapes proved positive for the presence of mites. Based on specific, morphological characteristics, the identified mites were classified as species of the *Demodex* genus.

The body sizes of 7 *Demodex* mites (3 males and 4 females) are presented in Table 1 and Figure 2.

The arithmetic average of the mites’ total length was: 198.33 µm, with 15 µm (gnathosoma), 55 µm (podosoma) and 127.66 µm (opisthosoma), for males; 201.5 µm, with 18 µm (gnathosoma), 49 µm (podosoma) and 134.5 µm (opisthosoma), in females.

Following detailed morphological examination, we established that the identified mites were *Demodex cati*.

Demodex species are prostigmatic mites with a body consisting of: gnathosoma, podosoma and opisthosoma, with four rudimentary, short, and telescopic legs that give them a vermiform appearance. *D. cati* has a trapezoidal gnathosoma, the width of the base exceeding the length. The opisthosoma is about five times longer than wide, representing about 2/3 of the total length. It is striated transversely and ends tapered [9].

According to studies, *D. cati* is similar in size to those identified in our study. Lowenstein et al. [10] noticed values of 181.7 ± 17.9 µm of the total length, in males, with the gnathosoma measuring 14.7 ± 1.2 µm, the podosoma 52.2 ± 2.6 µm and the opisthosoma 114.2 ± 17.0 µm. Females had a total length of 219.0 ± 27.4 µm, with the gnathosoma measuring 16.2 ± 1.9 µm, the podosoma 58.7 ± 2.6 µm and the opisthosoma 143.9 ± 25.4 µm. Values of 181.7 ± 17.9 µm (males) and 219 ± 27.4 µm (females), representing the total length were reported by Taffin et al. [7].

In the case of *D. gatoi*, studies indicate values of 90.6±4.8µm (gnathosoma 14.5 ± 0.4 µm, podosoma 39.5 ± 1.3 µm, opisthosoma 36.6 ± 4.1 µm) [3] and 90.6 ± 4.8 µm [7], representing the total length of males. The total length values noticed in females were 108.3 ± 4.4 µm (gnathosoma 15.5 ± 0.7 µm, podosoma 42.3 ± 1.6 µm, opisthosoma 50.3 ± 4.0 µm) [3] and 108.3 ± 4.4 µm [7].

Smaller sizes, respectively 170.8 ± 1.4 µm (gnathosoma 24.3 ± 0.8 µm, podosoma 58.2 ± 0.9 µm, opisthosoma 88.4 ± 0.5 µm) [8] and 140–175 µm [7], representing the total length, were noticed in *Demodex* spp. males.

PCR amplification of DNA extracted from *Demodex* mites produced a single band of ~330 bp. 

The product matched 100% with *D. cati* sequences deposited in GenBank JX193759 throughout the entire compared region.

The used primers are highly conserved among *Demodex* species, including *D. gatoi, D. caprae, D. brevis, D. folliculorum, D. canis, D. injai* and an unnamed species affecting cats [8,11].

The quick test (FIV Ab/FeLV Ag-Vet Expert) results came back positive for FIV Ab and negative for FeLV Ag. The feline immunodeficiency virus and feline leukemia virus (FIV and FeLV, respectively) are severe diseases that frequently affect domestic cats causing immunosuppression and consequently, predisposing felines to secondary diseases (such as, feline demodicosis) [7,12,13].

According to the antibiogram, the recommended antibiotic class to be used in this case was the fluoroquinolones. 

The applied therapy based on the use of fluralaner and moxidectin proved successful, and 3 weeks post-treatment the skin scrapes became negative, and the clinical aspect had improved considerably, as shown in Figure 3. Consequently, alopecia, erythema, erosions, ulcerations, crusts, pruritus, and self-trauma disappeared. 

Moreover, following the recommended treatment against bacterial otitis, dark, agglutinated cerumen was not present in the external ear canal.

The efficacy of fluralaner in feline demodicosis has been evaluated before [14,15], while moxidectin has also been used previously for the treatment of canine demodicosis [16,17], as well as in the control of various feline parasitic infections.

No adverse reactions were recorded following administration of the product. Moreover, the easy application of the product has avoided supplementary stress for the animal. No mites could be identified any longer within three weeks following the administration of the product, as well as during the entire 4-month follow-up period.

The clinical history, physical examination, and microscopic observation of the mite in skin scrapings are used to diagnose *Demodex* spp. infections, however molecular methods of detection, such as polymerase chain reaction (PCR), have substantially improved infection diagnosis [8,17].

We may thus conclude that this is the first molecular study to highlight the presence of demodicosis (caused by *Demodex cati*) within the feline population of Romania, and that therapy using fluralaner and moxidectin leads to the complete resolution of clinical signs in the studied patient.

## Figures and Tables

**Figure 1 pathogens-10-01474-f001:**
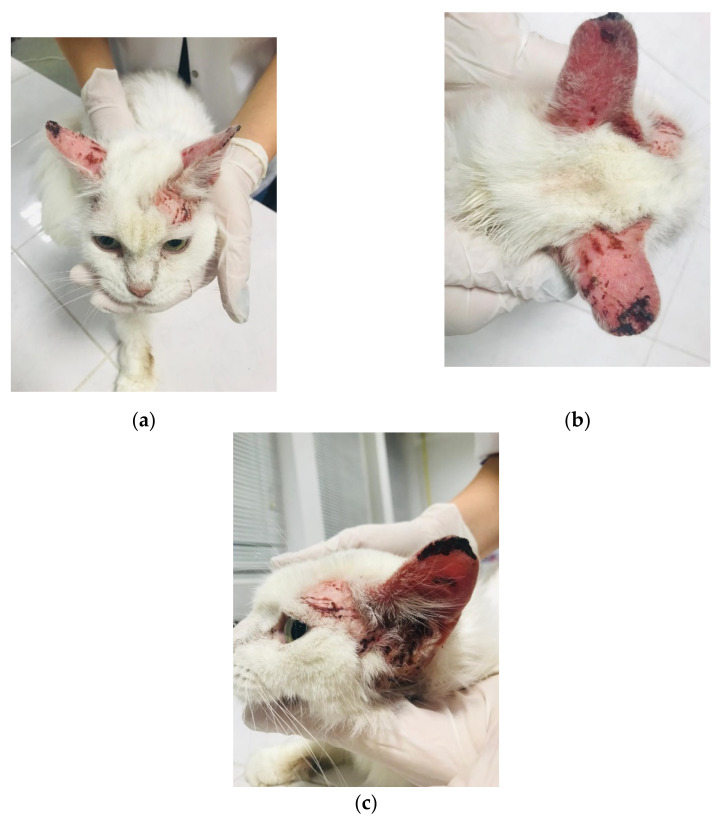
Feline demodicosis—lesional aspects: (**a**,**b**)—alopecia, erythema, small ulcerations and crusts in the external ear canal (before treatment); (**c**)—black secretions are present in the external ear canal.

**Figure 2 pathogens-10-01474-f002:**
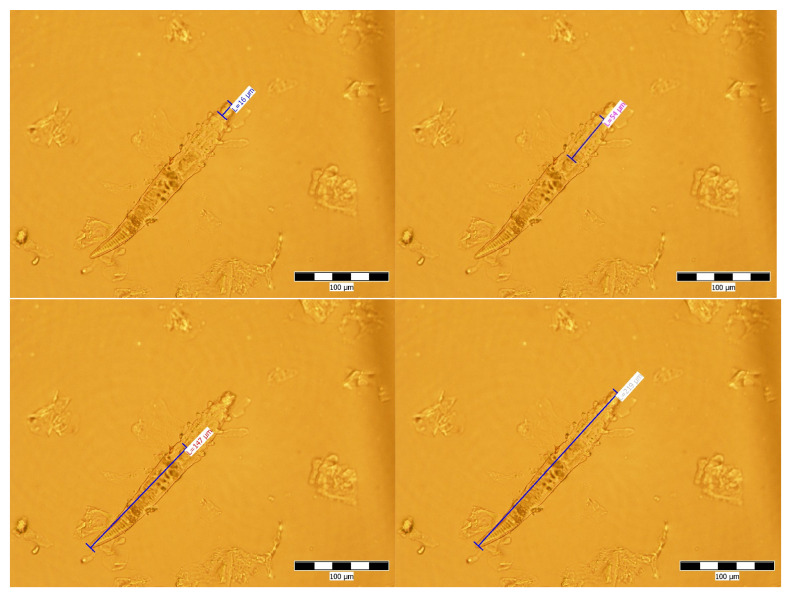
Male *Demodex*—body segments sizes.

**Figure 3 pathogens-10-01474-f003:**
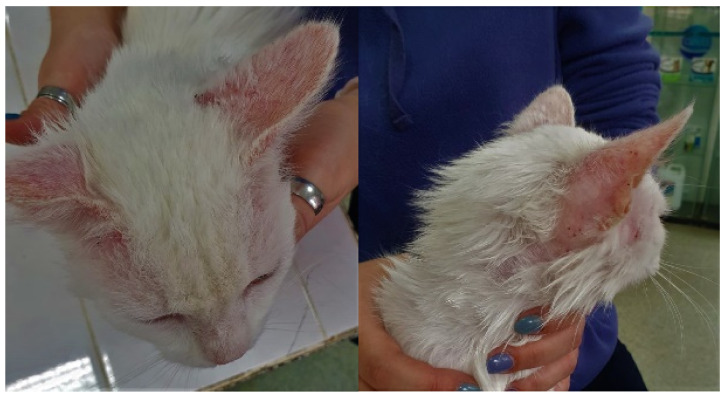
Hair regrowth in the ear area and decrease in size of ulcerative lesions, absence of crusts (following treatment, 3 weeks).

**Table 1 pathogens-10-01474-t001:** Demodex mites body sizes.

No.	Sex	Total Length	Gnathosoma	Podosoma	Opisthosoma
1.	Male	188 µm	13 µm	59 µm	116 µm
2.	Male	219 µm	16 µm	54 µm	147 µm
3.	Male	188 µm	16 µm	52 µm	120 µm
4.	Female	214 µm	26 µm	41 µm	147 µm
5.	Female	207 µm	18 µm	50 µm	139 µm
6.	Female	197 µm	12 µm	53 µm	132 µm
7.	Female	188 µm	16 µm	52 µm	120 µm

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
