# Peer review of "Feline Demodicosis Case Report—First Molecular Characterization of Demodex Mites in Romania"

_pathogens, 2021, doi:10.3390/pathogens10111474_

Round 1

Reviewer 1 Report

This is an interesting case description. I have few minor remarks for you:

Line 27-28: "(Acari: Demodecidae or Arachnida: Demodecidae)" - neither of them. Acari have been confirmed to be diphiletic (Acariformes and Parasitiformes) thus not a natural taxon. Of course the very name Acari traditionaly remains but I suggest you use the most correct form which is: Acariformes: Demodecidae.

Line 29: The second Demodex and each from this point (unless at the beginning of the sentence) should be abbreviated to D.

Line 32: What host specificity? High, I assume but it must be stated.

Figure 1. needs improved caption.

Lines 123-124: "Following detailed morphological examination, we established that the identified mites were Demodex cati." - based on which key? Please name at least the most important taxonomic traits of these mites. I am asking because not only veterinarians may read and use your work in the future. The more that you present a reference molecular data of a quite rare species.

As for references, Latin names of species should always be in italics, even in the titles of articles.

Author Response

Dear reviewer 1,

Thank you for taking the time to review our manuscript and we greatly appreciate the suggestions made to improve our paper: Feline Demodicosis Case Report—First Molecular Characterization of Demodex Mites in Romania - Pathogens-1454065

Revisions to the manuscript were marked up using the “Track Changes” option.

In the following we will respond to your comments and we will include the requested additional information.

Please see the pdf.

Sincerely!

The Authors

Reviewer 2 Report

The case report, despite being only one case, aggregates information about the infection by Demodex cati, as in terms of geographic distribution as response to the treatment. Two points must be changed:
1) Items 2.1.1 and 2.1.2 must be changed by a single item, as the history (2.1.1) is part of the clinical examination (2.1.2).
2) I suggest including the Genebank deposit number from the mite sequence.

Author Response

Dear reviewer 2,

Thank you for taking the time to review our manuscript and we greatly appreciate the suggestions made to improve our paper: Feline Demodicosis Case Report—First Molecular Characterization of Demodex Mites in Romania - Pathogens-1454065

Revisions to the manuscript were marked up using the “Track Changes” option.

In the following we will respond to your comments and we will include the requested additional information.

Please see the pdf.

Sincerely!

The Authors

Reviewer 3 Report

This case report is interesting and detailed, but I would suggest a few modifications to improve it.

Line 8 and line 28: would change “extremely rare” to uncommon to rare, according to the literature

Lines 10-11 and 52 -53: The authors state that alopecia and erythema were self inflicted, but they could have also been caused by the demodicosis. I would suggest to refrase it as: “Alopecia, erythema, minor erosions and ulcerations and crusts, associated with pruritus and self trauma were observed on the head”.

Line 31: Missing a ).

Lie 41: The title of the paragraph is “material and methods”, therefore you should not describe the results of your exams here. Maybe the titles of the paragraphs can be changed, as this is a case report, in “methods and results” and “discussion”. Alternatively, the authors should write the methods in this section, the the results in the following. This also applies elsewhere (e.g. line 78: the results of the FIV test..)

Lines 58 – 66: To follow a logical sequence I would suggest to list the differential diagnoses first. A possible rephrase would be: “The main differential diagnoses included a ringworm infection, Malassezia dermatitis, bacterial infections and mite infestations. We performed a wood lamp’s….

Line 59: Which bacteria were identified? Cocci or rods? Did you perform a quantitative/semi quantitave evaluation? This is quite important to interpret the results of the bacterial colture and antibiogram.

Line 64: Mineral oil does not clarify, I would rephrase as: Several deep skin scrapes were performed from the lesion margins, the obtained material was mixed in mineral oil, a cover-slip was applied and the samples were examined directly, using the 10x magnification objective of a microscope.

Lines 71 and following: an Antibiogram does not confirm the diagnosis, I would skin lined 71 and 72 and rephase as “Following otoscopy and cytological examination of the ear secretions, we ruled out Otodectes cynotis infestation and establish the diagnostic of bacterial otitis. The following bacterial strains were identified through the subsequent bacterial culture: Staphylococcus intermedius, Pseudomonas aeruginosa, beta-hemolytic Streptococci. An antibiogram was also performed, highlighting sensitivity of the identified pathogens (every one?) to chloramphenicol, ciprofloxacin, tobramycin, amikacin and resistance to trimethoprim, lincomycin, spectinomycin.

Line 110: as suggested above, the authors must decide if they want to report all the results here or in the previous section. If they choose to maintain the current paragraph division, they should report in this paragraph the results of all the diagnostics. Also, there is not a clear description on the clinical improvement in the main text, I would suggest stating it clearly, e.g.: on follow up visit on day 30, the cat was improved…

Lines 149  - 152: Fiv and FelV are known underlying causes for feline demodicosis, This phrase can be improved as it is not clear.

Lines 153-154: Would omit this, as the usefulness of an antibiogram in a case of external otitis with a mixed bacterial growth is questionable.

Line 172: Would change produced with caused by

Author Response

Dear reviewer 3,

Thank you for taking the time to review our manuscript and we greatly appreciate the suggestions made to improve our paper: Feline Demodicosis Case Report—First Molecular Characterization of Demodex Mites in Romania - Pathogens-1454065

Revisions to the manuscript were marked up using the “Track Changes” option.

In the following, we will respond to your comments and we will include the requested additional information.

In my manuscript (not everywhere) there is a slight discrepancy regarding the number of lines specified by you, but I have identified the problematic text in the suggestions you wrote.

Please see the pdf.

Sincerely!

The Authors
